# Adaptive Submodular Maximization in Bandit Setting

**Victor Gabillon**
INRIA Lille - team SequeL
Villeneuve d'Ascq, France
*victor.gabillon@inria.fr*

**Branislav Kveton**
Technicolor Labs
Palo Alto, CA
*branislav.kveton@technicolor.com*

**Zheng Wen**
Electrical Engineering Department
Stanford University
*zhengwen@stanford.edu*

**Brian Eriksson**
Technicolor Labs
Palo Alto, CA
*brian.eriksson@technicolor.com*

**S. Muthukrishnan**
Department of Computer Science
Rutgers
*muthu@cs.rutgers.edu*

## Abstract

Maximization of submodular functions has wide applications in machine learning and artificial intelligence. Adaptive submodular maximization has been traditionally studied under the assumption that the model of the world, the expected gain of choosing an item given previously selected items and their states, is known. In this paper, we study the setting where the expected gain is initially unknown, and it is learned by interacting repeatedly with the optimized function. We propose an efficient algorithm for solving our problem and prove that its expected cumulative regret increases logarithmically with time. Our regret bound captures the inherent property of submodular maximization, earlier mistakes are more costly than later ones. We refer to our approach as *Optimistic Adaptive Submodular Maximization (*OASM*)* because it trades off exploration and exploitation based on the *optimism in the face of uncertainty* principle. We evaluate our method on a preference elicitation problem and show that non-trivial $K$-step policies can be learned from just a few hundred interactions with the problem.

## 1   Introduction

Maximization of submodular functions [14] has wide applications in machine learning and artificial intelligence, such as social network analysis [9], sensor placement [10], and recommender systems [7, 2]. In this paper, we study the problem of adaptive submodular maximization [5]. This problem is a variant of submodular maximization where each item has a state and this state is revealed when the item is chosen. The goal is to learn a policy that maximizes the expected return for choosing $K$ items.

Adaptive submodular maximization has been traditionally studied in the setting where the model of the world, the expected gain of choosing an item given previously selected items and their states, is known. This is the first paper that studies the setting where the model is initially unknown, and it is learned by interacting repeatedly with the environment. We bring together the concepts of adaptive submodular maximization and bandits, and the result is an efficient solution to our problem.

We make four major contributions. First, we propose a model where the expected gain of choosing an item can be learned efficiently. The main assumption in the model is that the state of each item is distributed independently of the other states. Second, we propose *Optimistic Adaptive Submodular Maximization (*OASM*)*, a bandit algorithm that selects items with the highest upper confidence bound on the expected gain. This algorithm is computationally efficient and easy to implement. Third, we prove that the expected cumulative regret of our algorithm increases logarithmically with time. Our regret bound captures the inherent property of adaptive submodular maximization, earlier mistakes are more costly than later ones. Finally, we apply our approach to a real-world preference elicitation

problem and show that non-trivial policies can be learned from just a few hundred interactions with the problem.

## 2 Adaptive Submodularity

In adaptive submodular maximization, the objective is to maximize, under constraints, a function of the form:

$$f : 2^I \times \{-1, 1\}^L \to \mathbb{R}, \tag{1}$$

where $I = \{1, \ldots, L\}$ is a set of $L$ *items* and $2^I$ is its power set. The first argument of $f$ is a subset of *chosen items* $A \subseteq I$. The second argument is the *state* $\phi \in \{-1, 1\}^L$ of all items. The $i$-th entry of $\phi$, $\phi[i]$, is the state of item $i$. The state $\phi$ is drawn i.i.d. from some probability distribution $P(\Phi)$. The reward for choosing items $A$ in state $\phi$ is $f(A, \phi)$. For simplicity of exposition, we assume that $f(\emptyset, \phi) = 0$ in all $\phi$. In problems of our interest, the state is only partially observed. To capture this phenomenon, we introduce the notion of observations. An observation is a vector $\mathbf{y} \in \{-1, 0, 1\}^L$ whose non-zero entries are the observed states of items. We say that $\mathbf{y}$ is an *observation* of state $\phi$, and write $\phi \sim \mathbf{y}$, if $\mathbf{y}[i] = \phi[i]$ in all non-zero entries of $\mathbf{y}$. Alternatively, the state $\phi$ can be viewed as a *realization* of $\mathbf{y}$, one of many. We denote by $\mathrm{dom}(\mathbf{y}) = \{i : \mathbf{y}[i] \neq 0\}$ the *observed items* in $\mathbf{y}$ and by $\phi\langle A \rangle$ the observation of items $A$ in state $\phi$. We define a partial ordering on observations and write $\mathbf{y}' \succeq \mathbf{y}$ if $\mathbf{y}'[i] = \mathbf{y}[i]$ in all non-zero entries of $\mathbf{y}$, $\mathbf{y}'$ is a more specific observation than $\mathbf{y}$. In the terminology of Golovin and Krause [5], $\mathbf{y}$ is a *subrealization* of $\mathbf{y}'$.

We illustrate our notation on a simple example. Let $\phi = (1, 1, -1)$ be a state, and $\mathbf{y}_1 = (1, 0, 0)$ and $\mathbf{y}_2 = (1, 0, -1)$ be observations. Then all of the following claims are true:

$$\phi \sim \mathbf{y}_1, \quad \phi \sim \mathbf{y}_2, \quad \mathbf{y}_2 \succeq \mathbf{y}_1, \quad \mathrm{dom}(\mathbf{y}_2) = \{1, 3\}, \quad \phi\langle\{1, 3\}\rangle = \mathbf{y}_2, \quad \phi\langle\mathrm{dom}(\mathbf{y}_1)\rangle = \mathbf{y}_1.$$

Our goal is to maximize the expected value of $f$ by adaptively choosing $K$ items. This problem can be viewed as a $K$ step game, where at each step we choose an item according to some policy $\pi$ and then observe its state. A *policy* $\pi : \{-1, 0, 1\}^L \to I$ is a function from observations $\mathbf{y}$ to items. The observations represent our past decisions and their outcomes. A *$k$-step policy* in state $\phi$, $\pi_k(\phi)$, is a collection of the first $k$ items chosen by policy $\pi$. The policy is defined recursively as:

$$\pi_k(\phi) = \pi_{k-1}(\phi) \cup \left\{\pi_{[k]}(\phi)\right\}, \qquad \pi_{[k]}(\phi) = \pi(\phi\langle\pi_{k-1}(\phi)\rangle), \qquad \pi_0(\phi) = \emptyset, \tag{2}$$

where $\pi_{[k]}(\phi)$ is the $k$-th item chosen by policy $\pi$ in state $\phi$. The optimal $K$-step policy satisfies:

$$\pi^* = \arg\max_\pi \mathbb{E}_\phi[f(\pi_K(\phi), \phi)]. \tag{3}$$

In general, the problem of computing $\pi^*$ is NP-hard [14, 5]. However, near-optimal policies can be computed efficiently when the maximized function has a *diminishing return* property. Formally, we require that the function is adaptive submodular and adaptive monotonic [5].

**Definition 1.** *Function $f$ is* adaptive submodular *if:*

$$\mathbb{E}_\phi[f(A \cup \{i\}, \phi) - f(A, \phi) \mid \phi \sim \mathbf{y}_A] \geq \mathbb{E}_\phi[f(B \cup \{i\}, \phi) - f(B, \phi) \mid \phi \sim \mathbf{y}_B]$$

*for all items $i \in I \setminus B$ and observations $\mathbf{y}_B \succeq \mathbf{y}_A$, where $A = \mathrm{dom}(\mathbf{y}_A)$ and $B = \mathrm{dom}(\mathbf{y}_B)$.*

**Definition 2.** *Function $f$ is* adaptive monotonic *if $\mathbb{E}_\phi[f(A \cup \{i\}, \phi) - f(A, \phi) \mid \phi \sim \mathbf{y}_A] \geq 0$ for all items $i \in I \setminus A$ and observations $\mathbf{y}_A$, where $A = \mathrm{dom}(\mathbf{y}_A)$.*

In other words, the expected gain of choosing an item is always non-negative and does not increase as the observations become more specific.

Let $\pi^g$ be the *greedy policy* for maximizing $f$, a policy that always selects the item with the highest expected gain:

$$\pi^g(\mathbf{y}) = \arg\max_{i \in I \setminus \mathrm{dom}(\mathbf{y})} g_i(\mathbf{y}), \tag{4}$$

where:

$$g_i(\mathbf{y}) = \mathbb{E}_\phi[f(\mathrm{dom}(\mathbf{y}) \cup \{i\}, \phi) - f(\mathrm{dom}(\mathbf{y}), \phi) \mid \phi \sim \mathbf{y}] \tag{5}$$

is the *expected gain* of choosing item $i$ after observing $\mathbf{y}$. Then, based on the result of Golovin and Krause [5], $\pi^g$ is a $(1 - 1/e)$-approximation to $\pi^*$, $\mathbb{E}_\phi[f(\pi_K^g(\phi), \phi)] \geq (1 - 1/e)\mathbb{E}_\phi[f(\pi_K^*(\phi), \phi)]$, if $f$ is adaptive submodular and adaptive monotonic. In the rest of this paper, we say that an observation $\mathbf{y}$ is a *context* if it can be observed under the greedy policy $\pi^g$. Specifically, there exist $k$ and $\phi$ such that $\mathbf{y} = \phi\langle\pi_k^g(\phi)\rangle$.

# 3 Adaptive Submodularity in Bandit Setting

The greedy policy $\pi^g$ can be computed only if the objective function $f$ and the distribution of states $P(\Phi)$ are known, because both of these quantities are needed to compute the marginal benefit $g_i(\mathbf{y})$ (Equation 5). In practice, the distribution $P(\Phi)$ is often unknown, for instance in a newly deployed sensor network where the failure rates of the sensors are unknown. In this paper, we study a natural variant of adaptive submodular maximization that can model such problems. The distribution $P(\Phi)$ is assumed to be unknown and we learn it by interacting repeatedly with the problem.

## 3.1 Model

The problem of learning $P(\Phi)$ can be cast in many ways. One approach is to directly learn the joint $P(\Phi)$. This approach is not practical for two reasons. First, the number of states $\phi$ is exponential in the number of items $L$. Second, the state of our problem is observed only partially. As a result, it is generally impossible to identify the distribution that generates $\phi$. Another possibility is to learn the probability of individual states $\phi[i]$ conditioned on context, observations $\mathbf{y}$ under the greedy policy $\pi^g$ in up to $K$ steps. This is impractical because the number of contexts is exponential in $K$.

Clearly, additional structural assumptions are necessary to obtain a practical solution. In this paper, we assume that the states of items are independent of the context in which the items are chosen. In particular, the state $\phi[i]$ of each item $i$ is drawn i.i.d. from a Bernoulli distribution with mean $p_i$. In this setting, the joint probability distribution factors as:

$$P(\Phi = \phi) = \prod_{i=1}^{L} p_i^{\mathbb{1}\{\phi[i]=1\}} (1 - p_i)^{1-\mathbb{1}\{\phi[i]=1\}} \tag{6}$$

and the problem of learning $P(\Phi)$ reduces to estimating $L$ parameters, the means of the Bernoullis. A major question is how restrictive is our independence assumption. We argue that this assumption is fairly natural in many applications. For instance, consider a sensor network where the sensors fail at random due to manufacturing defects. The failures of these sensors are independent of each other and thus can be modeled in our framework. To validate our assumption, we conduct an experiment (Section 4) that shows that it does not greatly affect the performance of our method on a real-world problem. Correlations obviously exist and we discuss how to model them in Section 6.

Based on the independence assumption, we rewrite the expected gain (Equation 5) as:

$$g_i(\mathbf{y}) = p_i \bar{g}_i(\mathbf{y}), \tag{7}$$

where:

$$\bar{g}_i(\mathbf{y}) = \mathbb{E}_\phi[\, f(\mathrm{dom}(\mathbf{y}) \cup \{i\}, \phi) - f(\mathrm{dom}(\mathbf{y}), \phi) \mid \phi \sim \mathbf{y}, \phi[i] = 1 \,] \tag{8}$$

is the expected gain when item $i$ is in state 1. For simplicity of exposition, we assume that the gain is zero when the item is in state $-1$. We discuss how to relax this assumption in Appendix.

In general, the gain $\bar{g}_i(\mathbf{y})$ depends on $P(\Phi)$ and thus cannot be computed when $P(\Phi)$ is unknown. In this paper, we assume that $\bar{g}_i(\mathbf{y})$ can be computed without knowing $P(\Phi)$. This scenario is quite common in practice. In maximum coverage problems, for instance, it is quite reasonable to assume that the covered area is only a function of the chosen items and their states. In other words, the gain can be computed as $\bar{g}_i(\mathbf{y}) = f(\mathrm{dom}(\mathbf{y}) \cup \{i\}, \phi) - f(\mathrm{dom}(\mathbf{y}), \phi)$, where $\phi$ is any state such that $\phi \sim \mathbf{y}$ and $\phi[i] = 1$.

Our learning problem comprises $n$ episodes. In episode $t$, we adaptively choose $K$ items according to some policy $\pi^t$, which may differ from episode to episode. The quality of the policy is measured by the expected cumulative $K$-step return $\mathbb{E}_{\phi_1,\ldots,\phi_n}[\sum_{t=1}^{n} f(\pi_K^t(\phi_t), \phi_t)]$. We compare this return to that of the greedy policy $\pi^g$ and measure the difference between the two returns by the *expected cumulative regret*:

$$R(n) = \mathbb{E}_{\phi_1,\ldots,\phi_n}\left[\sum_{t=1}^{n} R_t(\phi_t)\right] = \mathbb{E}_{\phi_1,\ldots,\phi_n}\left[\sum_{t=1}^{n} f(\pi_K^g(\phi_t), \phi_t) - f(\pi_K^t(\phi_t), \phi_t)\right]. \tag{9}$$

In maximum coverage problems, the greedy policy $\pi^g$ is a good surrogate for the optimal policy $\pi^*$ because it is a $(1 - 1/e)$-approximation to $\pi^*$ (Section 2).

---

**Algorithm 1** OASM: Optimistic adaptive submodular maximization.

---

**Input:** States $\phi_1, \ldots, \phi_n$

**for all** $i \in I$ **do** Select item $i$ and set $\hat{p}_{i,1}$ to its state, $T_i(0) \leftarrow 1$ **end for**   $\triangleright$ Initialization
**for all** $t = 1, 2, \ldots, n$ **do**
    $A \leftarrow \emptyset$
    **for all** $k = 1, 2, \ldots, K$ **do**   $\triangleright$ $K$-step maximization
        $\mathbf{y} \leftarrow \phi_t \langle A \rangle$
        $A \leftarrow A \cup \left\{ \arg\max\limits_{i \in I \setminus A} (\hat{p}_{i,T_i(t-1)} + c_{t-1,T_i(t-1)}) \bar{g}_i(\mathbf{y}) \right\}$   $\triangleright$ Choose the highest index
    **end for**
    **for all** $i \in I$ **do** $T_i(t) \leftarrow T_i(t-1)$ **end for**   $\triangleright$ Update statistics
    **for all** $i \in A$ **do**
        $T_i(t) \leftarrow T_i(t) + 1$
        $\hat{p}_{i,T_i(t)} \leftarrow \frac{1}{T_i(t)}(\hat{p}_{i,T_i(t-1)}T_i(t-1) + \frac{1}{2}(\phi_t[i] + 1))$
    **end for**
**end for**

---

## 3.2 Algorithm

Our algorithm is designed based on the *optimism in the face of uncertainty* principle, a strategy that is at the core of many bandit algorithms [1, 8, 13]. More specifically, it is a greedy policy where the expected gain $g_i(\mathbf{y})$ (Equation 7) is substituted for its optimistic estimate. The algorithm adaptively maximizes a submodular function in an optimistic fashion and therefore we refer to it as *Optimistic Adaptive Submodular Maximization (*OASM*)*.

The pseudocode of our method is given in Algorithm 1. In each episode, we maximize the function $f$ in $K$ steps. At each step, we compute the *index* $(\hat{p}_{i,T_i(t-1)} + c_{t-1,T_i(t-1)}) \bar{g}_i(\mathbf{y})$ of each item that has not been selected yet and then choose the item with the highest index. The terms $\hat{p}_{i,T_i(t-1)}$ and $c_{t-1,T_i(t-1)}$ are the maximum-likelihood estimate of the probability $p_i$ from the first $t - 1$ episodes and the radius of the confidence interval around this estimate, respectively. Formally:

$$\hat{p}_{i,s} = \frac{1}{s} \sum_{z=1}^{s} \frac{1}{2}(\phi_{\tau(i,z)}[i] + 1), \qquad c_{t,s} = \sqrt{\frac{2 \log(t)}{s}}, \tag{10}$$

where $s$ is the number of times that item $i$ is chosen and $\tau(i, z)$ is the index of the episode in which item $i$ is chosen for the $z$-th time. In episode $t$, we set $s$ to $T_i(t - 1)$, the number of times that item $i$ is selected in the first $t - 1$ episodes. The radius $c_{t,s}$ is designed such that each index is with high probability an upper bound on the corresponding gain. The index enforces exploration of items that have not been chosen very often. As the number of past episodes increases, all confidence intervals shrink and our method starts exploiting most profitable items. The $\log(t)$ term guarantees that each item is explored infinitely often as $t \to \infty$, to avoid linear regret.

Algorithm OASM has several notable properties. First, it is a greedy method. Therefore, our policies can be computed very fast. Second, it is guaranteed to behave near optimally as our estimates of the gain $g_i(\mathbf{y})$ become more accurate. We prove this claim in Section 3.3. Finally, our algorithm learns only $L$ parameters and therefore is quite practical. Specifically, note that if an item is chosen in one context, it helps in refining the estimate of the gain $g_i(\mathbf{y})$ in all other contexts.

## 3.3 Analysis

In this section, we prove an upper bound on the expected cumulative regret of Algorithm OASM in $n$ episodes. Before we present the main result, we define notation used in our analysis. We denote by $i^*(\mathbf{y}) = \pi^g(\mathbf{y})$ the item chosen by the greedy policy $\pi^g$ in context $\mathbf{y}$. Without loss of generality, we assume that this item is unique in all contexts. The hardness of discriminating between items $i$ and $i^*(\mathbf{y})$ is measured by a gap between the expected gains of the items:

$$\Delta_i(\mathbf{y}) = g_{i^*(\mathbf{y})}(\mathbf{y}) - g_i(\mathbf{y}). \tag{11}$$

Our analysis is based on counting how many times the policies $\pi^t$ and $\pi^g$ choose a different item at step $k$. Therefore, we define several variables that describe the state of our problem at this step. We

denote by $\mathcal{Y}_k(\pi) = \bigcup_\phi \{\phi\langle\pi_{k-1}(\phi)\rangle\}$ the set of all possible observations after policy $\pi$ is executed for $k-1$ steps. We write $\mathcal{Y}_k = \mathcal{Y}_k(\pi^g)$ and $\mathcal{Y}_k^t = \mathcal{Y}_k(\pi^t)$ when we refer to the policies $\pi^g$ and $\pi^t$, respectively. Finally, we denote by $\mathcal{Y}_{k,i} = \mathcal{Y}_k \cap \{\mathbf{y} : i \neq i^*(\mathbf{y})\}$ the set of contexts where item $i$ is suboptimal at step $k$.

Our main result is Theorem 1. Supplementary material for its proof is in Appendix. The terms *item* and *arm* are treated as synonyms, and we use whichever is more appropriate in a given context.

**Theorem 1.** *The expected cumulative regret of Algorithm* OASM *is bounded as:*

$$R(n) \leq \underbrace{\sum_{i=1}^{L} \ell_i \sum_{k=1}^{K} G_k \alpha_{i,k}}_{O(\log n)} + \underbrace{\frac{2}{3}\pi^2 L(L+1) \sum_{k=1}^{K} G_k}_{O(1)}, \tag{12}$$

*where $G_k = (K - k + 1) \max_{\mathbf{y} \in \mathcal{Y}_k} \max_i g_i(\mathbf{y})$ is an upper bound on the expected gain of the policy $\pi^g$*

*from step $k$ forward, $\ell_{i,k} = \left\lceil 8 \max_{\mathbf{y} \in \mathcal{Y}_{k,i}} \frac{\bar{g}_i^2(\mathbf{y})}{\Delta_i^2(\mathbf{y})} \log n \right\rceil$ is the number of pulls after which arm $i$ is not*

*likely to be pulled suboptimally at step $k$, $\ell_i = \max_k \ell_{i,k}$, and $\alpha_{i,k} = \frac{1}{\ell_i}\left[\ell_{i,k} - \max_{k' < k} \ell_{i,k'}\right]^+ \in [0,1]$*

*is a weight that associates the regret of arm $i$ to step $k$ such that $\sum_{k=1}^{K} \alpha_{i,k} = 1$.*

*Proof.* Our theorem is proved in three steps. First, we associate the regret in episode $t$ with the first step where our policy $\pi^t$ selects a different item from the greedy policy $\pi^g$. For simplicity, suppose that this step is step $k$. Then the regret in episode $t$ can be written as:

$$R_t(\phi_t) = f(\pi_K^g(\phi_t), \phi_t) - f(\pi_K^t(\phi_t), \phi_t)$$
$$= \underbrace{f(\pi_K^g(\phi_t), \phi_t) - f(\pi_{k-1}^g(\phi_t), \phi_t)}_{F_{k\to}^g(\phi_t)} - \underbrace{[f(\pi_K^t(\phi_t), \phi_t) - f(\pi_{k-1}^t(\phi_t), \phi_t)]}_{F_{k\to}^t(\phi_t)}, \tag{13}$$

where the last equality is due to the assumption that $\pi_{[j]}^t(\phi_t) = \pi_{[j]}^g(\phi_t)$ for all $j < k$; and $F_{k\to}^g(\phi_t)$ and $F_{k\to}^t(\phi_t)$ are the gains of the policies $\pi^g$ and $\pi^t$, respectively, in state $\phi_t$ from step $k$ forward. In practice, the first step where the policies $\pi^t$ and $\pi^g$ choose a different item is unknown, because $\pi^g$ is unknown. In this case, the regret can be written as:

$$R_t(\phi_t) = \sum_{i=1}^{L} \sum_{k=1}^{K} \mathbb{1}_{i,k,t}(\phi_t)(F_{k\to}^g(\phi_t) - F_{k\to}^t(\phi_t)), \tag{14}$$

where:

$$\mathbb{1}_{i,k,t}(\phi) = \mathbb{1}\left\{ \left(\forall j < k : \pi_{[j]}^t(\phi) = \pi_{[j]}^g(\phi)\right), \pi_{[k]}^t(\phi) \neq \pi_{[k]}^g(\phi), \pi_{[k]}^t(\phi) = i \right\} \tag{15}$$

is the indicator of the event that the policies $\pi^t$ and $\pi^g$ choose the same first $k-1$ items in state $\phi$, disagree in the $k$-th item, and $i$ is the $k$-th item chosen by $\pi^t$. The commas in the indicator function represent logical conjunction.

Second, in Lemma 1 we bound the expected loss associated with choosing the first different item at step $k$ by the probability of this event and an upper bound on the expected loss $G_k$, which does not depend on $\pi^t$ and $\phi_t$. Based on this result, we bound the expected cumulative regret as:

$$\mathbb{E}_{\phi_1,\dots,\phi_n}\left[\sum_{t=1}^{n} R_t(\phi_t)\right] = \mathbb{E}_{\phi_1,\dots,\phi_n}\left[\sum_{t=1}^{n}\sum_{i=1}^{L}\sum_{k=1}^{K} \mathbb{1}_{i,k,t}(\phi_t)(F_{k\to}^g(\phi_t) - F_{k\to}^t(\phi_t))\right]$$

$$= \sum_{i=1}^{L}\sum_{k=1}^{K}\sum_{t=1}^{n} \mathbb{E}_{\phi_1,\dots,\phi_{t-1}}\left[\mathbb{E}_{\phi_t}\left[\mathbb{1}_{i,k,t}(\phi_t)(F_{k\to}^g(\phi_t) - F_{k\to}^t(\phi_t))\right]\right]$$

$$\leq \sum_{i=1}^{L}\sum_{k=1}^{K}\sum_{t=1}^{n} \mathbb{E}_{\phi_1,\dots,\phi_{t-1}}\left[\mathbb{E}_{\phi_t}\left[\mathbb{1}_{i,k,t}(\phi_t)\right] G_k\right]$$

$$= \sum_{i=1}^{L}\sum_{k=1}^{K} G_k \mathbb{E}_{\phi_1,\dots,\phi_n}\left[\sum_{t=1}^{n} \mathbb{1}_{i,k,t}(\phi_t)\right]. \tag{16}$$

Finally, motivated by the analysis of UCB1 [1], we rewrite the indicator $\mathbb{1}_{i,k,t}(\phi_t)$ as:

$$\mathbb{1}_{i,k,t}(\phi_t) = \mathbb{1}_{i,k,t}(\phi_t)\mathbb{1}\{T_i(t-1) \leq \ell_{i,k}\} + \mathbb{1}_{i,k,t}(\phi_t)\mathbb{1}\{T_i(t-1) > \ell_{i,k}\}, \qquad (17)$$

where $\ell_{i,k}$ is a problem-specific constant. In Lemma 4, we show how to choose $\ell_{i,k}$ such that arm $i$ at step $k$ is pulled suboptimally a constant number of times in expectation after $\ell_{i,k}$ pulls. Based on this result, the regret corresponding to the events $\mathbb{1}\{T_i(t-1) > \ell_{i,k}\}$ is bounded as:

$$\sum_{i=1}^{L}\sum_{k=1}^{K} G_k \mathbb{E}_{\phi_1,\ldots,\phi_n}\left[\sum_{t=1}^{n}\mathbb{1}_{i,k,t}(\phi_t)\mathbb{1}\{T_i(t-1) > \ell_{i,k}\}\right] \leq \frac{2}{3}\pi^2 L(L+1)\sum_{k=1}^{K} G_k. \qquad (18)$$

On the other hand, the regret associated with the events $\mathbb{1}\{T_i(t-1) \leq \ell_{i,k}\}$ is trivially bounded by $\sum_{i=1}^{L}\sum_{k=1}^{K} G_k \ell_{i,k}$. A tighter upper bound is proved below:

$$\sum_{i=1}^{L}\mathbb{E}_{\phi_1,\ldots,\phi_n}\left[\sum_{k=1}^{K} G_k \sum_{t=1}^{n}\mathbb{1}_{i,k,t}(\phi_t)\mathbb{1}\{T_i(t-1) \leq \ell_{i,k}\}\right]$$

$$\leq \sum_{i=1}^{L}\max_{\phi_1,\ldots,\phi_n}\left[\sum_{k=1}^{K} G_k \sum_{t=1}^{n}\mathbb{1}_{i,k,t}(\phi_t)\mathbb{1}\{T_i(t-1) \leq \ell_{i,k}\}\right]$$

$$\leq \sum_{i=1}^{L}\sum_{k=1}^{K} G_k \left[\ell_{i,k} - \max_{k'<k}\ell_{i,k'}\right]^{+}. \qquad (19)$$

The last inequality can be proved as follows. Our upper bound on the expected loss at step $k$, $G_k$, is monotonically decreasing with $k$, and therefore $G_1 \geq G_2 \geq \ldots \geq G_K$. So for any given arm $i$, the highest cumulative regret subject to the constraint $T_i(t-1) \leq \ell_{i,k}$ at step $k$ is achieved as follows. The first $\ell_{i,1}$ mistakes are made at the first step, $[\ell_{i,2} - \ell_{i,1}]^{+}$ mistakes are made at the second step, $[\ell_{i,3} - \max\{\ell_{i,1}, \ell_{i,2}\}]^{+}$ mistakes are made at the third step, and so on. Specifically, the number of mistakes at step $k$ is $[\ell_{i,k} - \max_{k'<k}\ell_{i,k'}]^{+}$ and the associated loss is $G_k$.

Our main claim follows from combining the upper bounds in Equations 18 and 19. ∎

### 3.4 Discussion of Theoretical Results

Algorithm OASM mimics the greedy policy $\pi^g$. Therefore, we decided to prove Theorem 1 based on counting how many times the policies $\pi^t$ and $\pi^g$ choose a different item. Our proof has three parts. First, we associate the regret in episode $t$ with the first step where the policy $\pi^t$ chooses a different item from $\pi^g$. Second, we bound the expected regret in each episode by the probability of deviating from the policy $\pi^g$ at step $k$ and an upper bound on the associated loss $G_k$, which depends only on $k$. Finally, we divide the expected cumulative regret into two terms, before and after item $i$ at step $k$ is selected a sufficient number of times $\ell_{i,k}$, and then set $\ell_{i,k}$ such that both terms are $O(\log n)$. We would like to stress that our proof is relatively general. Our modeling assumptions (Section 3.1) are leveraged only in Lemma 4. In the rest of the proof, we only assume that $f$ is adaptive submodular and adaptive monotonic.

Our regret bound has several notable properties. First, it is logarithmic in the number of episodes $n$, through problem-specific constants $\ell_{i,k}$. So we recover a classical result from the bandit literature. Second, the bound is polynomial in all constants of interest, such as the number of items $L$ and the number of maximization steps $K$ in each episode. We would like to stress that it is not linear in the number of contexts $\mathcal{Y}_K$ at step $K$, which is exponential in $K$. Finally, note that our bound captures the shape of the optimized function $f$. In particular, because the function $f$ is adaptive submodular, the upper bound on the gain of the policy $\pi^g$ from step $k$ forward, $G_k$, decreases as $k$ increases. As a result, earlier deviations from $\pi^g$ are penalized more than later ones.

## 4 Experiments

Our algorithm is evaluated on a preference elicitation problem in a movie recommendation domain. This problem is cast as asking $K$ yes-or-no movie-genre questions. The users and their preferences are extracted from the *MovieLens* dataset [11], a dataset of 6k users who rated one million movies.

| Genre | $g_i(\mathbf{0})$ | $\bar{g}_i(\mathbf{0})$ | $P(\phi[i]=1)$ |
|---|---|---|---|
| Crime | 4.1% | 13.0% | 0.32 |
| Children's | 4.1% | 9.2% | 0.44 |
| Animation | 3.2% | 6.6% | 0.48 |
| Horror | 3.0% | 8.0% | 0.38 |
| Sci-Fi | 2.8% | 23.0% | 0.12 |
| Musical | 2.6% | 6.0% | 0.44 |
| Fantasy | 2.6% | 5.8% | 0.44 |
| Adventure | 2.3% | 19.6% | 0.12 |

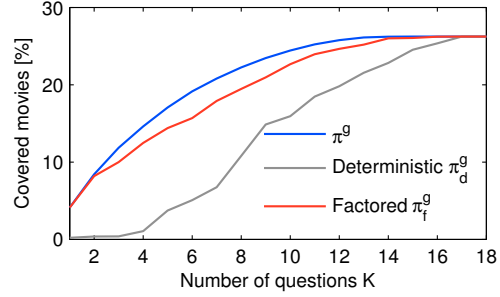

Figure 1: **Left**. Eight movie genres that cover the largest number of movies in expectation. **Right**. Comparison of three greedy policies for solving our preference elicitation problem. For each policy and $K \leq L$, we report the expected percentage of covered movies after $K$ questions.

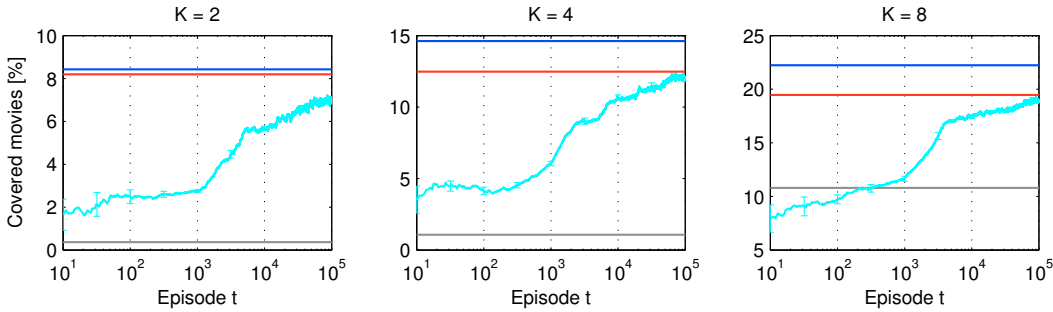

Figure 2: The expected return of the `OASM` policy $\pi^t$ (cyan lines) in all episodes up to $t = 10^5$. The return is compared to those of the greedy policies $\pi^g$ (blue lines), $\pi^g_f$ (red lines), and $\pi^g_d$ (gray lines) in the offline setting (Figure 1) at the same operating point, the number of asked questions $K$.

We choose 500 most rated movies from the dataset. Each movie $l$ is represented by a feature vector $\mathbf{x}_l$ such that $\mathbf{x}_l[i] = 1$ if the movie belongs to genre $i$ and $\mathbf{x}_l[i] = 0$ if it does not. The preference of user $j$ for genre $i$ is measured by tf-idf, a popular importance score in information retrieval [12]. In particular, it is defined as $\text{tf-idf}(j,i) = \#(j,i) \log\left(\frac{n_u}{\#(\cdot,i)}\right)$, where $\#(j,i)$ is the number of movies from genre $i$ rated by user $j$, $n_u$ is the number of users, and $\#(\cdot,i)$ is the number of users that rated at least one movie from genre $i$. Intuitively, this score prefers genres that are often rated by the user but rarely rated overall. Each user $j$ is represented by a genre preference vector $\phi$ such that $\phi[i] = 1$ when genre $i$ is among five most favorite genres of the user. These genres cover on average 25% of our movies. In Figure 1, we show several popular genres from our dataset.

The reward for asking user $\phi$ questions $A$ is:

$$f(A, \phi) = \tfrac{1}{5} \sum_{l=1}^{500} \max_i \left[ \mathbf{x}_l[i] \mathbb{1}\{\phi[i] = 1\} \mathbb{1}\{i \in A\} \right], \tag{20}$$

the percentage of movies that belong to at least one genre $i$ that is preferred by the user and queried in $A$. The function $f$ captures the notion that knowing more preferred genres is better than knowing less. It is submodular in $A$ for any given preference vector $\phi$, and therefore adaptive submodular in $A$ when the preferences are distributed independently of each other (Equation 6). In this setting, the expected value of $f$ can be maximized near optimally by a greedy policy (Equation 4).

In the first experiment, we show that our assumption on $P(\Phi)$ (Equation 6) is not very restrictive in our domain. We compare three greedy policies for maximizing $f$ that know $P(\Phi)$ and differ in how the expected gain of choosing items is estimated. The first policy $\pi^g$ makes no assumption on $P(\Phi)$ and computes the gain according to Equation 5. The second policy $\pi^g_f$ assumes that the distribution $P(\Phi)$ is factored and computes the gain using Equation 7. Finally, the third policy $\pi^g_d$ computes the gain according to Equation 8, essentially ignoring the stochasticity of our problem. All policies are applied to all users in our dataset for all $K \leq L$ and their expected returns are reported in Figure 1. We observe two trends. First, the policy $\pi^g_f$ usually outperforms the policy $\pi^g_d$ by a large margin. So although our independence assumption may be incorrect, it is a better approximation than ignoring

the stochastic nature of the problem. Second, the expected return of $\pi_{\mathrm{f}}^g$ is always within 84% of $\pi^g$. We conclude that $\pi_{\mathrm{f}}^g$ is a good approximation to $\pi^g$.

In the second experiment, we study how the OASM policy $\pi^t$ improves over time. In each episode $t$, we randomly choose a new user $\phi_t$ and then the policy $\pi^t$ asks $K$ questions. The expected return of $\pi^t$ is compared to two offline baselines, $\pi_{\mathrm{f}}^g$ and $\pi_{\mathrm{d}}^g$. The policies $\pi_{\mathrm{f}}^g$ and $\pi_{\mathrm{d}}^g$ can be viewed as upper and lower bounds on the expected return of $\pi^t$, respectively. Our results are shown in Figure 2. We observe two major trends. First, $\pi^t$ easily outperforms the baseline $\pi_{\mathrm{d}}^g$ that ignores the stochasticity of our problem. In two cases, this happens in less than ten episodes. Second, the expected return of $\pi^t$ approaches that of $\pi_{\mathrm{f}}^g$, as is expected based on our analysis.

## 5 Related Work

Our paper is motivated by prior work in the areas of submodularity [14, 5] and bandits [1]. Similar problems to ours were studied by several authors. For instance, Yue and Guestrin [17], and Guillory and Bilmes [6], applied bandits to submodular problems in a non-adaptive setting. In our work, we focus on the adaptive setting. This setting is more challenging because we learn a $K$-step policy for choosing items, as opposing to a single set of items. Wen *et al.* [16] studied a variant of generalized binary search, *sequential Bayesian search*, where the policy for asking questions is learned on-the-fly by interacting with the environment. A major observation of Wen *et al.* [16] is that this problem can be solved near optimally without exploring. As a result, its solution and analysis are completely different from those in our paper.

Learning with trees was studied in machine learning in many settings, such as online learning with tree experts [3]. This work is similar to ours only in trying to learn a tree. The notions of regret and the assumptions on solved problems are completely different. *Optimism in the face of uncertainty* is a popular approach to designing learning algorithms, and it was previously applied to more general problems than ours, such as planning [13] and MDPs [8]. Both of these solutions are impractical in our setting. The former assumes that the model of the world is known and the latter is computationally intractable.

## 6 Conclusions

This is the first work that studies adaptive submodular maximization in the setting where the model of the world is initially unknown. We propose an efficient bandit algorithm for solving the problem and prove that its expected cumulative regret increases logarithmically with time. Our work can be viewed as reinforcement learning (RL) [15] for adaptive submodularity. The main difference in our setting is that we can learn near-optimal policies without estimating the value function. Learning of value functions is typically hard, even when the model of the problem is known. Fortunately, this is not necessary in our problem and therefore we can develop a very efficient learning algorithm.

We assume that the states of items are distributed independently of each other. In our experiments, this assumption was less restrictive than we expected (Section 4). Nevertheless, we believe that our approach should be studied under less restrictive assumptions. In preference elicitation (Section 4), for instance, the answers to questions are likely to be correlated due to many factors, such as user's preferences, user's mood, and the similarity of the questions. Our current model cannot capture any of these dependencies. However, we believe that our approach is quite general and can be extended to more complex models. We think that any such generalization would comprise three major steps: choosing a model of $P(\Phi)$, deriving a corresponding upper confidence bound on the expected gain, and finally proving an equivalent of Lemma 4.

We also assume that the expected gain of choosing an item (Equation 7) can be written as a product of some known gain function (Equation 8) and the probability of the item's states. This assumption is quite natural in maximum coverage problems but may not be appropriate in other problems, such as generalized binary search [4].

Our upper bound on the expected regret at step $k$ (Lemma 1) may be loose in practice because it is obtained by maximizing over all contexts $\mathbf{y} \in \mathcal{Y}_k$. In general, it is difficult to prove a tighter bound. Such a bound would have to depend on the probability of making a mistake in a specific context at step $k$, which depends on the policy in that episode, and indirectly on the progress of learning in all earlier episodes. We leave this for future work.

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
