[Supplementary Material]

## A Technical Lemmas

**Lemma 1.** *For all episodes $t$ and $k \leq K$:*

$$\mathbb{E}_{\phi_t}\big[\mathbb{1}_{i,k,t}(\phi_t)(F^g_{k\to}(\phi_t) - F^t_{k\to}(\phi_t))\big] \leq \mathbb{E}_{\phi_t}[\mathbb{1}_{i,k,t}(\phi_t)]\, G_k.$$

*Proof.* The first term in the expectation can be written as:

$$\mathbb{E}_{\phi_t}[\mathbb{1}_{i,k,t}(\phi_t)F^g_{k\to}(\phi_t)] \stackrel{(a)}{=} \sum_{\mathbf{y}\in\mathcal{Y}_k} P(\phi_t \sim \mathbf{y})\mathbb{E}_{\phi_t}\big[\mathbb{1}_{i,k,t}(\phi_t)F^g_{k\to}(\phi_t)\,|\,\phi_t \sim \mathbf{y}\big]$$

$$\stackrel{(b)}{=} \sum_{\mathbf{y}\in\mathcal{Y}_k} P(\phi_t \sim \mathbf{y})\mathbb{1}_{i,k,t}(\mathbf{y})\mathbb{E}_{\phi_t}\big[F^g_{k\to}(\phi_t)\,|\,\phi_t \sim \mathbf{y}\big]. \qquad (21)$$

The equality (a) is due to conditioning $\phi_t$ on the first $k-1$ observations $\mathbf{y}$ under the policy $\pi^g$. The equality (b) follows from the fact that the indicator $\mathbb{1}_{i,k,t}(\phi_t)$ (Equation 15) does not depend on the observations from step $k$ forward. The last two terms in Equation 21 may depend on each other in a non-trivial manner. To break this dependency, we first bound the expectation over $\mathbf{y}$ using Hölder's inequality and then the latter term by Lemma 2:

$$\mathbb{E}_{\phi_t}[\mathbb{1}_{i,k,t}(\phi_t)F^g_{k\to}(\phi_t)] \leq \mathbb{E}_{\phi_t}[\mathbb{1}_{i,k,t}(\phi_t)] \max_{\mathbf{y}\in\mathcal{Y}_k} \mathbb{E}_{\phi_t}\big[F^g_{k\to}(\phi_t)\,|\,\phi_t \sim \mathbf{y}\big]$$

$$\leq \mathbb{E}_{\phi_t}[\mathbb{1}_{i,k,t}(\phi_t)]\, G_k, \qquad (22)$$

where $G_k = (K-k+1)\max_{\mathbf{y}\in\mathcal{Y}_k}\max_i g_i(\mathbf{y})$ is an upper bound on the expected gain of $\pi^g$ from step $k$ forward. The term $G_k$ is independent of $\phi_t$ and $\pi_t$, the state and policy in episode $t$.

Similarly to Equation 21, the second term in the expectation can be written as:

$$\mathbb{E}_{\phi_t}\big[\mathbb{1}_{i,k,t}(\phi_t)F^t_{k\to}(\phi_t)\big] = \sum_{\mathbf{y}\in\mathcal{Y}^t_k} P(\phi_t \sim \mathbf{y})\mathbb{1}_{i,k,t}(\mathbf{y})\mathbb{E}_{\phi_t}\big[F^t_{k\to}(\phi_t)\,|\,\phi_t \sim \mathbf{y}\big]. \qquad (23)$$

In Lemma 3, we show that $\mathbb{E}_{\phi_t}\big[F^t_{k\to}(\phi_t)\,|\,\phi_t \sim \mathbf{y}\big] \geq 0$ for all $\mathbf{y}$ and $k$. It follows that:

$$-\mathbb{E}_{\phi_t}\big[\mathbb{1}_{i,k,t}(\phi_t)F^t_{k\to}(\phi_t)\big] \leq 0. \qquad (24)$$

Our main claim is obtained by combining the upper bounds in Equations 22 and 24. $\blacksquare$

**Lemma 2.** *For all $k \leq K$:*

$$\max_{\mathbf{y}\in\mathcal{Y}_k} \mathbb{E}_{\phi}\big[F^g_{k\to}(\phi)\,|\,\phi \sim \mathbf{y}\big] \leq (K-k+1)\max_{\mathbf{y}\in\mathcal{Y}_k}\max_i g_i(\mathbf{y}).$$

*Proof.* First, we note that for all contexts $\mathbf{y}\in\mathcal{Y}_k$:

$$\mathbb{E}_{\phi}\big[F^g_{k\to}(\phi)\,|\,\phi \sim \mathbf{y}\big]$$

$$= \sum_{j=k}^{K} \mathbb{E}_{\phi}\big[f(\pi^g_j(\phi),\phi) - f(\pi^g_{j-1}(\phi),\phi)\,|\,\phi \sim \mathbf{y}\big]$$

$$= \sum_{j=k}^{K}\sum_{\mathbf{y}'\in\mathcal{Y}_j} P(\phi \sim \mathbf{y}'\,|\,\phi \sim \mathbf{y})\mathbb{E}_{\phi}\big[f(\pi^g_j(\phi),\phi) - f(\pi^g_{j-1}(\phi),\phi)\,|\,\phi \sim \mathbf{y}, \phi \sim \mathbf{y}'\big]$$

$$= \sum_{j=k}^{K}\sum_{\mathbf{y}'\in\mathcal{Y}_j} P(\phi \sim \mathbf{y}'\,|\,\phi \sim \mathbf{y})\mathbb{E}_{\phi}\big[f(\pi^g_j(\phi),\phi) - f(\pi^g_{j-1}(\phi),\phi)\,|\,\phi \sim \mathbf{y}'\big]. \qquad (25)$$

The last equality follows from the fact that $P(\phi \sim \mathbf{y}'\,|\,\phi \sim \mathbf{y}) > 0$ implies $\mathbf{y}' \succeq \mathbf{y}$, and this further implies that $(\phi \sim \mathbf{y}) \wedge (\phi \sim \mathbf{y}') \equiv \phi \sim \mathbf{y}'$. Second, because the policy $\pi^g$ in context $\mathbf{y}$ chooses an item with the largest expected gain and $f$ is adaptive submodular, we know that:

$$\mathbb{E}_{\phi}\big[f(\pi^g_j(\phi),\phi) - f(\pi^g_{j-1}(\phi),\phi)\,|\,\phi \sim \mathbf{y}'\big] \leq \max_i g_i(\mathbf{y}) \qquad (26)$$

for all $j$ and $\mathbf{y}' \succeq \mathbf{y}$. This upper bound can be substituted into Equation 25 and yields:

$$\mathbb{E}_{\phi}\big[F^g_{k\to}(\phi)\,|\,\phi \sim \mathbf{y}\big] \leq (K-k+1)\max_i g_i(\mathbf{y}). \qquad (27)$$

Our main claim is obtained by maximizing both sides of the above inequality over $\mathbf{y}$. $\blacksquare$

**Lemma 3.** *For all $k \leq K$ and $\mathbf{y} \in \mathcal{Y}_k^t$:*

$$\mathbb{E}_\phi\big[\, F_{k\to}^t(\phi)\,\big|\,\phi \sim \mathbf{y}\,\big] \geq 0.$$

*Proof.* Similarly to Lemma 2, we note that for all $\mathbf{y} \in \mathcal{Y}_k^t$:

$$\mathbb{E}_\phi\big[\, F_{k\to}^t(\phi)\,\big|\,\phi \sim \mathbf{y}\,\big]$$

$$= \sum_{j=k}^{K} \mathbb{E}_\phi\big[\, f(\pi_j^t(\phi), \phi) - f(\pi_{j-1}^t(\phi), \phi)\,\big|\,\phi \sim \mathbf{y}\,\big]$$

$$= \sum_{j=k}^{K} \sum_{\mathbf{y}' \in \mathcal{Y}_j^t} P(\phi \sim \mathbf{y}' \mid \phi \sim \mathbf{y}) \mathbb{E}_\phi\big[\, f(\pi_j^t(\phi), \phi) - f(\pi_{j-1}^t(\phi), \phi)\,\big|\,\phi \sim \mathbf{y}, \phi \sim \mathbf{y}'\,\big]$$

$$= \sum_{j=k}^{K} \sum_{\mathbf{y}' \in \mathcal{Y}_j^t} P(\phi \sim \mathbf{y}' \mid \phi \sim \mathbf{y}) \mathbb{E}_\phi\big[\, f(\pi_j^t(\phi), \phi) - f(\pi_{j-1}^t(\phi), \phi)\,\big|\,\phi \sim \mathbf{y}'\,\big]. \tag{28}$$

The last equality follows from the fact that $P(\phi \sim \mathbf{y}' \mid \phi \sim \mathbf{y}) > 0$ implies $\mathbf{y}' \succeq \mathbf{y}$, and this further implies that $(\phi \sim \mathbf{y}) \wedge (\phi \sim \mathbf{y}') \equiv \phi \sim \mathbf{y}'$. Because $f$ is adaptive monotonic, we know that:

$$\mathbb{E}_\phi\big[\, f(\pi_j^t(\phi), \phi) - f(\pi_{j-1}^t(\phi), \phi)\,\big|\,\phi \sim \mathbf{y}'\,\big] \geq 0 \tag{29}$$

for all $j$ and $\mathbf{y}'$. Our main claim follows from substituting the above bound into Equation 28. ∎

**Lemma 4.** *For all arms $i$ and $k \leq K$:*

$$\mathbb{E}_{\phi_1, \ldots, \phi_n}\left[ \sum_{t=1}^{n} \mathbb{1}_{i,k,t}(\phi_t) \mathbb{1}\{T_i(t-1) > \ell_{i,k}\} \right] \leq \frac{2}{3}\pi^2(L+1), \tag{30}$$

*where* $\ell_{i,k} = \left\lceil 8 \max_{\mathbf{y} \in \mathcal{Y}_{k,i}} \frac{\bar{g}_i^2(\mathbf{y})}{\Delta_i^2(\mathbf{y})} \log n \right\rceil$.

*Proof.* Our proof has the same structure as the proof of Theorem 1 by Auer *et al.* [1]. Let $\ell_{i,k}$ be a positive integer. Then for all arms $i$ and steps $k$:

$$\sum_{t=1}^{n} \mathbb{1}_{i,k,t}(\phi_t) \mathbb{1}\{T_i(t-1) > \ell_{i,k}\}$$

$$= \sum_{t=\ell_{i,k}+1}^{n} \mathbb{1}_{i,k,t}(\phi_t) \mathbb{1}\{T_i(t-1) > \ell_{i,k}\}$$

$$\leq \sum_{t=\ell_{i,k}+1}^{n} \mathbb{1}\big\{\exists \mathbf{y} \in \mathcal{Y}_{k,i} : (\hat{p}_{i,T_i(t-1)} + c_{t-1,T_i(t-1)})\bar{g}_i(\mathbf{y}) \geq$$

$$(\hat{p}_{i^*(\mathbf{y}),T_{i^*(\mathbf{y})}(t-1)} + c_{t-1,T_{i^*(\mathbf{y})}(t-1)})\bar{g}_{i^*(\mathbf{y})}(\mathbf{y}), \; T_i(t-1) > \ell_{i,k}\big\}$$

$$\leq \sum_{t=\ell_{i,k}+1}^{n} \sum_{s=1}^{t} \sum_{s_i=\ell_{i,k}+1}^{t} \mathbb{1}\big\{\exists \mathbf{y} \in \mathcal{Y}_{k,i} : (\hat{p}_{i,s_i} + c_{t-1,s_i})\bar{g}_i(\mathbf{y}) \geq (\hat{p}_{i^*(\mathbf{y}),s} + c_{t-1,s})\bar{g}_{i^*(\mathbf{y})}(\mathbf{y})\big\}$$

$$= \sum_{t=\ell_{i,k}}^{n-1} \sum_{s=1}^{t+1} \sum_{s_i=\ell_{i,k}+1}^{t+1} \mathbb{1}\big\{\exists \mathbf{y} \in \mathcal{Y}_{k,i} : (\hat{p}_{i,s_i} + c_{t,s_i})\bar{g}_i(\mathbf{y}) \geq (\hat{p}_{i^*(\mathbf{y}),s} + c_{t,s})\bar{g}_{i^*(\mathbf{y})}(\mathbf{y})\big\}. \tag{31}$$

The existence of $\mathbf{y} \in \mathcal{Y}_{k,i}$ such that $(\hat{p}_{i,s_i} + c_{t,s_i})\bar{g}_i(\mathbf{y}) \geq (\hat{p}_{i^*(\mathbf{y}),s} + c_{t,s})\bar{g}_{i^*(\mathbf{y})}(\mathbf{y})$ implies that at least one of the following claims must be true:

$$\exists \mathbf{y} \in \mathcal{Y}_{k,i} : \qquad \hat{p}_{i^*(\mathbf{y}),s} \leq p_{i^*(\mathbf{y})} - c_{t,s} \tag{32}$$

$$\hat{p}_{i,s_i} \geq p_i + c_{t,s_i} \tag{33}$$

$$\exists \mathbf{y} \in \mathcal{Y}_{k,i} : \quad p_{i^*(\mathbf{y})}\bar{g}_{i^*(\mathbf{y})}(\mathbf{y}) < p_i \bar{g}_i(\mathbf{y}) + 2c_{t,s_i}\bar{g}_i(\mathbf{y}). \tag{34}$$

We bound the probability of the first two events (Equations 32 and 33) using Hoeffding's inequality and the union bound:

$$P(\exists \mathbf{y} \in \mathcal{Y}_{k,i} : \hat{p}_{i^*(\mathbf{y}),s} \leq p_{i^*(\mathbf{y})} - c_{t,s}) \leq L \exp[-4 \log t] = L t^{-4} \tag{35}$$

$$P(\hat{p}_{i,s_i} \geq p_i + c_{t,s_i}) \leq \exp[-4 \log t] = t^{-4}. \tag{36}$$

When $\ell_{i,k} = \left\lceil 8 \max_{\mathbf{y} \in \mathcal{Y}_{k,i}} \frac{\bar{g}_i^2(\mathbf{y})}{\Delta_i^2(\mathbf{y})} \log n \right\rceil$, the third event (Equation 34) cannot happen. In particular, for all $\mathbf{y} \in \mathcal{Y}_{k,i}$ and $s_i \geq 8 \max_{\mathbf{y} \in \mathcal{Y}_{k,i}} \frac{\bar{g}_i^2(\mathbf{y})}{\Delta_i^2(\mathbf{y})} \log n$, we can show that:

$$
\begin{aligned}
p_{i^*(\mathbf{y})} \bar{g}_{i^*(\mathbf{y})}(\mathbf{y}) - p_i \bar{g}_i(\mathbf{y}) - 2c_{t,s_i} \bar{g}_i(\mathbf{y}) &= \bar{g}_i(\mathbf{y}) \left[ \frac{\Delta_i(\mathbf{y})}{\bar{g}_i(\mathbf{y})} - 2\sqrt{\frac{2 \log t}{s_i}} \right] \\
&\geq \bar{g}_i(\mathbf{y}) \left[ \frac{\Delta_i(\mathbf{y})}{\bar{g}_i(\mathbf{y})} - \min_{\mathbf{y} \in \mathcal{Y}_{k,i}} \frac{\Delta_i(\mathbf{y})}{\bar{g}_i(\mathbf{y})} \right] \\
&\geq 0. \tag{37}
\end{aligned}
$$

Therefore, we may conclude that:

$$
\begin{aligned}
\mathbb{E}_{\phi_1,\ldots,\phi_n} &\left[ \sum_{t=1}^n \mathbb{1}_{i,k,t}(\phi_t) \mathbb{1}\{T_i(t-1) > \ell_{i,k}\} \right] \\
&\leq \sum_{t=1}^\infty \sum_{s=1}^{t+1} \sum_{s_i=1}^{t+1} \left[ P(\exists \mathbf{y} \in \mathcal{Y}_{k,i} : \hat{p}_{i^*(\mathbf{y}),s} \leq p_{i^*(\mathbf{y})} - c_{t,s}) + P(\hat{p}_{i,s_i} \geq p_i + c_{t,s_i}) \right] \\
&\leq (L+1) \sum_{t=1}^\infty (t+1)^2 t^{-4} \\
&\leq (L+1) \sum_{t=1}^\infty 4t^{-2} \\
&= \frac{2}{3} \pi^2 (L+1). \tag{38}
\end{aligned}
$$

∎

## B  Categorical State Variables

In this section, we show how to generalize our work to categorical state variables.

We assume that each item $i$ has $M$ possible states, $\phi[i] \in \{1, \ldots, M\}$. The state of item $i$ is drawn i.i.d. from a categorical distribution, which is described by $M$ probabilities $p_{i,1}, \ldots, p_{i,M}$ such that $\sum_{m=1}^M p_{i,m} = 1$. In this setting, the joint probability distribution of states factors as:

$$P(\Phi = \phi) = \prod_{i=1}^L \prod_{m=1}^M p_{i,m}^{\mathbb{1}\{\phi[i]=m\}}. \tag{39}$$

Based on the above assumption, we rewrite the expected gain (Equation 5) as:

$$g_i(\mathbf{y}) = \sum_{m=1}^M p_{i,m} \bar{g}_{i,m}(\mathbf{y}), \tag{40}$$

where:

$$\bar{g}_{i,m}(\mathbf{y}) = \mathbb{E}_\phi[f(\mathrm{dom}(\mathbf{y}) \cup \{i\}, \phi) - f(\mathrm{dom}(\mathbf{y}), \phi) \mid \phi \sim \mathbf{y}, \phi[i] = m] \tag{41}$$

is the expected gain when item $i$ is in state $m$. Similarly to Section 3.1, we assume that the function $\bar{g}_{i,m}(\mathbf{y})$ is known and can be computed without knowing $P(\Phi)$.

Algorithm `OASM` changes in the computation of the index. The index is computed as:

$$\sum_{m=1}^{M} (\hat{p}_{i,m,T_i(t-1)} + c_{t-1,T_i(t-1)})\bar{g}_{i,m}(\mathbf{y}), \tag{42}$$

where $\hat{p}_{i,m,T_i(t-1)}$ is the maximum-likelihood estimate of $p_{i,m}$ from the first $t-1$ episodes, which is computed from $T_i(t-1)$ observations of item $i$.

Our analysis changes in Lemma 4. First, the events in Equations 32 and 33 have to be bounded for all $m \in \{1, \dots, M\}$. Second, the event in Equation 34 does not happen when:

$$\ell_{i,k} = \left\lceil 8 \max_{\mathbf{y} \in \mathcal{Y}_{k,i}} \frac{\left(\sum_{m=1}^{M} \bar{g}_{i,m}(\mathbf{y})\right)^2}{\Delta_i^2(\mathbf{y})} \log n \right\rceil. \tag{43}$$

As a result, our final regret bound is:

$$R(n) \leq \underbrace{\sum_{i=1}^{L} \ell_i \sum_{k=1}^{K} G_k \alpha_{i,k}}_{O(\log n)} + \underbrace{\frac{2}{3}\pi^2 ML(L+1)\sum_{k=1}^{K} G_k}_{O(1)}. \tag{44}$$