[Reviews · NeurIPS 2013]

Submitted by Assigned_Reviewer_4

This paper extends the concept of adaptive submodularity (Golovin & Krause, JAIR 2011) to the
setting where one wants to maximize an adaptive submodular function whose form is not known,
but where the necessary training episodes are provided.

What differentiates adaptive submodularity for submodularity is that the set-valued function being
optimized is stochastic: the reward for selecting set A \subset L can vary with the state/context \phi \in {0,1}^L. We assume that \phi is drawn from a distribution P(\Phi). Call the reward f(A,\phi). A training episode is a sequence of steps, where the forecaster selects a set of items according to a policy. The policy accounts for the uncertainty in \phi.

The key ideas in this paper, which connect adaptive submodularity to multi-armed bandits are, in my opinion

K1. Learn the incremental gains, not the submodular function

The critical assumption made is that the reward of adding an element to a set does not depend on
the unknown state \phi, so one can generalize the reward gained from adding an element to a set across states \phi.

K2. Optimism in the face of uncertainty

Once you assume the incremental gains do not depend on the context, learning can be framed as a bandit problem where each element in the ground set corresponds to an arm. Under K1, the authors can frame learning in terms of the UCB1 algorithm on Bernoulli bandits, drawing heavily on the original UCB1 paper.

It is worth mentioning to the reader that the no-regret result in this paper depends heavily on K1 and K2.If one looks at the literature on learning monotone submodular functions in a more general setting (Balcan & Harvey, STOC 2011), the picture is much more dismal.

The proof of Theorem 1 is quite elegant, but the paper is definitely hampered by the tight page limitations of a NIPS submission. A reader who has not read Golovin & Krause will have only the foggiest idea of why this problem is interesting. One way of improving the clarity of this submission, in its own right, is to strip the redundant first paragraph in the conclusions, and use the space for exposition on how submodularity fits into this problem from a MAB perspective.

The paper clearly builds on two established ideas: adaptive submodularity and the UCB1 algorithm for multi-armed bandits. The clever bit is K1, which connects the two ideas.

The experiments (Section 4) are perfunctory, and seem pretty contrived. The ground set is the 19 genres. It's not at all clear what the training episodes are. It does seem though that the independence of gains and contexts is the same as assuming that a genres of a movie don't depend on the preferred genres of the users. But, I'd expect that there are more movies in more popular genres. Is there something that I am missing?
Summary: A nice connection between learning adaptive monotone, adaptive submodular functions and multi-armed bandits. The proof is clever, but the experimental section is a conceptual mess.

Submitted by Assigned_Reviewer_6

In adaptive submodular maximization, the world state is drawn from a known distribution, and induces a state for each item in a given set. The objective is to adaptively select K items in order to maximize a known function of both the subset of items chosen and the (initially unknown) world state, given that the state of an item is revealed immediately once it is selected. The authors consider a Bandit version of the problem in which the objective function is known, but the distribution from which the world state is sampled is not known. Instead, the algorithm plays several rounds (called episodes) in which the world state is sampled from a fixed but unknown distribution.

To make the problem more tractable, the authors make an independence assumption, such that the unknown distribution induces a product distribution on the states of the items. They also restrict themselves to binary item states {0,1}, and furthermore to objectives such that items in state 0 contribution no objective value (though this last assumption can be removed.) In this case, the authors propose a rather natural approach: an Upper-Confidence-Bound (UCB) style algorithm, which greedily selects elements with maximum optimistic estimates of expected marginal benefit.

The authors prove an O(log n) regret bound for this problem over n episodes, using an analysis that builds on that for UCB1. Their bound has some problem-dependent constants that are hard to interpret, and may be loose by up to a factor of K, but aside from that is near optimal. The algorithm is fairly easy to implement, and the authors empirically evaluate it on a preference elicitation task based on MovieLens data. They compare their algorithm (OASM) to an adaptive greedy baseline (with knowledge of the true distribution over world states), to a non-adaptive baseline, and to an adaptive greedy baseline run on a product-distribution of states which approximates the empirical distribution. They show that for this application, the product-distribution assumption is not terrible when comparing to adaptive greedy baseline, and that OASM approaches the performance of the former (beating the deterministic baseline after a few thousand episodes).

Other comments:

There are a few places where you are missing the second argument to f, such as the last instance in equation 14.

Perhaps you can take advantage of the bounds on approximate implementations of the adaptive greedy algorithm in [3] to refine your analysis?

How about non-binary state spaces? It seems like your analysis should generalize without too much trouble.
Summary: I like this paper. It has nice algorithmic contributions resulting in a simple, practical algorithm with nice theoretical bounds. It is well written overall, though I would like to see some more intuition about the bound in Theorem 1.

Submitted by Assigned_Reviewer_7

This paper addresses the problem of maximizing an adaptive submodular function in a bandit setting where the probability distribution of the states is unknown. The authors show that the accumulative regret bound increases logarithmically with time.

The paper is well written. However, it can be improved. For instance no intuition (or very little) is given for OASM. The explanation of the proof can also be improved. For instance, a road map can be provided before authors directly jump into details.

Pros:

This is the first regret bound for adaptive submodular maximization. I believe the contribution is significant.

Cons:

The results are based on two simplified assumptions. First the states assumed to be binary which is not generally true. Second the joint probability distribution assumed to be independent.

Summary: I think this is a nice paper and should be accepted.
Author Feedback

Author rebuttal: We thank all three reviewers for insightful feedback. We appreciate that all of you found the paper worthy of an accept recommendation with comments on how the paper is well-written, the proposed method is practical, and its analysis is elegant.

Clarity of presentation
***********************

We agree that the current version of the paper is sometimes too dense. In the next version of the paper, we will move the proof of the main theorem into the appendix and substitute it in the paper for a short sketch. The remaining space will be used to 1) explain the connection that we make between MABs and adaptive submodular maximization, 2) better motivate Algorithm OASM, 3) discuss problem-specific constants from Theorem 1 in detail, and 4) better describe our experimental setup.

Non-binary observations
***********************

Our approach and analysis can be extended to both 1) categorical item states and 2) the problems where the marginal contribution of choosing an item in any state is non-zero. We will note on these extensions in the next version of the paper.

Reviewer 4
**********

We regret that our experimental setup is not clear. The setup is as follows. In episode t, we elicit preferences of a randomly chosen user from our dataset, which corresponds to asking K yes-or-no movie genre questions. The preference elicitation policy in episode t is learned from interactions with the past t - 1 users, through the probabilities of answers to our questions. We make the following independence assumption. The probability that the user answers "yes" to the k-th question in episode t is independent of the user's past k - 1 questions and answers in that episode. This is equivalent to assuming that the profile of the user, answers to all questions, is a binary vector of length L, whose each entry is drawn iid from a Bernoulli distribution of an unknown mean before the episode starts. The profile does not change as the user answers questions. These clarifications will be included in the next version of the paper.